# HPV E6/E7 mRNA Testing in the Follow-Up of HPV-Vaccinated Patients After Treatment for High-Grade Cervical Intraepithelial Neoplasia

**DOI:** 10.3390/vaccines13080823

**Published:** 2025-07-31

**Authors:** Adolfo Loayza, Alicia Hernandez, Ana M. Rodriguez, Belen Lopez, Cristina Gonzalez, David Hardisson, Itziar de la Pena, Maria Serrano, Rocio Arnedo, Ignacio Zapardiel

**Affiliations:** 1Department of Gynecology and Obstetrics, La Paz University Hospital, 28046 Madrid, Spain; adolfomusi@gmail.com (A.L.); ahernandezg@salud.madrid.org (A.H.); belenlopezcavanillas@gmail.com (B.L.); crisgb2404@hotmail.com (C.G.); mserranovelasco@gmail.com (M.S.); rocioarnedo94@gmail.com (R.A.); 2Department of Pathology, La Paz University Hospital, 28046 Madrid, Spain; anamargarita.rodriguez@salud.madrid.org (A.M.R.); david.hardisson@salud.madrid.org (D.H.); itziar.delapena@salud.madrid.org (I.d.l.P.); 3Gynecologic Oncology Unit, La Paz University Hospital, 28046 Madrid, Spain

**Keywords:** cervical intraepithelial neoplasia, conization, human papillomavirus, E6 and E7 mRNA, HPV vaccination

## Abstract

**Introduction:** Following up on treated high-grade cervical intraepithelial neoplasia (HSIL/CIN) lesions poses a challenge. Cervical cytology often has a high false-negative rate, while high-risk human papillomavirus (HR-HPV) DNA testing, though sensitive, lacks specificity. The detection of messenger RNA of the HR-HPV E6 and E7 oncoproteins (E6/E7 mRNA) is proposed as an indicator of viral integration, which is crucial for identifying severe lesions. Additionally, HPV vaccination could reduce recurrence rates in patients treated for high-grade cervical intraepithelial neoplasia. **Objective:** Our study aimed to assess the clinical utility of E6/E7 mRNA determination in the follow-up of HPV-immunized patients who were treated for HSIL/CIN. **Methods:** We conducted a retrospective observational study including 407 patients treated for HSIL/CIN. The recurrence rate and the validity parameters of E6/E7 mRNA testing were analyzed. **Results:** The recurrence rate for high-grade lesions was 1.7%. This low percentage might be related to the vaccination of patients who were not immunized before treatment. The sensitivity of the E6/E7 mRNA test was 88% at the first clinical visit, reaching 100% in the second and third reviews. Specificity was 91% at the first visit, 92% at the second, and 85% at the third. Regarding predictive values, the positive predictive value was 18% at the first visit, 10% at the second, and 14% at the third, while the negative predictive value was 100% across all follow-up visits. **Conclusions:** The E6/E7 mRNA test appears to be an effective tool for ruling out recurrence after treatment for HSIL/CIN lesions in HPV-immunized patients.

## 1. Introduction

High-grade cervical intraepithelial neoplasia (HSIL/CIN) is the precursor lesion of cervical cancer (CC). Human papillomavirus (HPV) is responsible for 98.7% of CC [1]. Non-vaccinated HPV patients treated for pre-invasive lesions of the cervix have an elevated risk (5% to 30%) of having a persistence or recurrence of the lesion and, therefore, an increased risk of invasive cancer [2], although information in this regard is heterogeneous. In general, persistent disease is considered when HPV is detected at the first check-up after treatment (during the first 6–12 months); however, recurrent disease is defined when it re-appears after the first normal check-up.

Currently, the follow-up tests after treatment for HSIL/CIN include cervical cytology, HPV testing, and colposcopy. Cervical cytology has a low sensitivity (62%), which limits its effectiveness [3]. HPV DNA testing detects the L1 region, the structural protein of the virus. DNA testing detects the virus in patients who present a high prevalence, such as young women. It is known that most infections are transient, without clinical significance. This would explain the low specificity (75%) of the DNA test for the detection of HSIL/CIN or cancer [4]. In severe lesions, the entire viral genome or fragments thereof are integrated into the chromosomal DNA of the host cell. The most important consequence of this process is the overexpression of the HPV oncoproteins E6 and E7. These play a crucial role in carcinogenesis since they lead to increased genomic instability, acquisition of oncogenic mutations, and eventual malignant transformation [5]. The detection of E6 and E7 messenger RNA (E6/E7 mRNA) of high-risk HPV (HR-HPV) is an indicator, not only of infection, but also of viral integration, allowing the identification of more severe lesions. Follow-up studies in patients with low-risk cervical lesions (LSIL/CIN) conclude that the E6/E7 mRNA test has a higher specificity and the same sensitivity compared to the DNA test, allowing greater accuracy in the selection of women at risk of LSIL/CIN progression [6,7].

In response to growing scientific evidence, since 2015, our center has adopted the messenger RNA (mRNA) human papillomavirus (HPV) detection test as the primary tool for the control and follow-up of patients treated for high-grade cervical intraepithelial neoplasia (CIN). This change represented a significant evolution from previous methodologies, aligning us with the latest guidelines in cervical lesion screening and management. For this reason, we set out to conduct a review to determine the efficacy of this decision and confirm whether the implementation of the HPV mRNA detection test has resulted in a tangible improvement in the management and prognosis of our patients.

Furthermore, a free vaccination program has been implemented for all patients who, despite having been treated for high-grade cervical intraepithelial neoplasia (CIN), had not been previously immunized against the human papillomavirus (HPV).

Our study aimed to analyze recurrence rates and determine the clinical utility of the HR-HPV E6/E7 mRNA test in the follow-up of patients treated for high-grade cervical intraepithelial neoplasia (HSIL/CIN) who had been immunized against the human papillomavirus (HPV), either before or after treatment.

## 2. Materials and Methods

After obtaining Institutional Review Board (IRB) approval, a retrospective observational study was conducted to evaluate the clinical utility of diagnostic tests based on the detection of HPV mRNA. We reviewed 407 non-vaccinated HPV patients treated for CIN grade 2–3 between June 2015 and June 2018. At our center, loop electrosurgical excision procedure (LEEP) is the standard treatment for high-grade cervical lesions (CIN 2/CIN 3). This method allows for tissue removal and subsequent histological analysis. However, to minimize potential side effects, we perform laser vaporization in selected cases. This ablative technique is reserved for young patients who have completed their reproductive desires, lesions that are completely visible via colposcopy, and cases where there is no suspicion of invasive disease, as confirmed by prior biopsies. All patients included in the study were over 18 years of age. As part of the post-treatment follow-up protocol, all patients underwent HR-HPV E6/E7 mRNA testing at each clinical check-up. Additionally, free HPV vaccination was prescribed for those patients who had not been previously immunized against HPV.

Data on the following variables were collected: age of the patient at the time of diagnosis, cytological and histological diagnosis, therapeutic procedure, pathological outcome, margins in case of conization or LEEP (loop electrosurgical excision procedure), number of control visits, follow-up time, and results of the tests performed at each control visit. Our current follow-up protocol for patients treated for high-grade cervical intraepithelial neoplasia (CIN) involves an initial control visit scheduled at 4 months if surgical margins are involved, and at 6 months if margins are clear. This initial assessment is then followed by three subsequent annual controls. After these yearly evaluations, patients transition to biennial controls. A cervical cytology, HPV mRNA test, and colposcopy, with directed biopsy if required, were performed at each follow-up visit.

Treatment failure was defined as the presence of residual or recurrent high-grade lesion confirmed by biopsy at the first clinical visit.

A complete gynecological examination was performed at each control. The liquid-based cervical cytology sample was taken with both endocervical brush and spatula. The sample was processed by the Pathology Department. The fixed sample was processed in ThinPrep 5000 (ThinPrep Hologic, Bedford, MA, USA), following the manufacturer’s recommendations. The presence of viral RNA was evaluated using the Aptima HPV Assay (Hologic Iberia S.L., Madrid, Spain). It is a qualitative molecular test for the detection of 14 HPV genotypes (16/18/31/33/35/39/45/51/52/56/58/59/66/68) that later allows differentiation between types 16 and 18/45, following the manufacturer’s recommendations. In the processing of cervical biopsies, the samples, fixed in 10% buffered formalin, were embedded in paraffin and cut into 4-micron sections and stained with hematoxylin–eosin.

Qualitative data were described using absolute frequencies and percentages. Quantitative data were described using mean ± standard deviation. Comparison between groups of qualitative variables was performed with the Chi-square test. The usefulness of the mRNA test was estimated by calculating the sensitivity, specificity, and predictive values, with their corresponding 95% confidence intervals. The alpha error was set at 5%. Statistical analyses were performed with the SPSS v15 software (IBM Corp., Armouk, NY, USA).

## 3. Results

A total of 407 patients were included. Among them, 401 (97.4%) were women with a histological diagnosis of CIN, distributed as 297 (73%) with CIN 2 and 104 (25.6%) with CIN 3. Six patients (2.6%) were included who did not have a histological diagnosis prior to the intervention but presented with HSIL cytology, and a high-grade lesion was subsequently confirmed in the conization specimen. The mean age was 37.6 ± 9.08 years. The mean follow-up time of the patients was 3.78 ± 1.87 years.

The treatment performed was conization in 362 (88.9%) cases. A total of 253 (69.8%) were performed due to CIN 2, 103 (28.4%) due to CIN 3, and 6 (1.6%) with no previous histological diagnosis. Laser vaporization was performed as treatment in 45 (11.1%) women. Among them, 43 (95.6%) were for CIN 2 and 2 (4.4%) for CIN 3.

The histological results of the conization specimen were CIN 2 in 220 (60.7%) patients, CIN 3 in 96 (26.6%), and no residual lesion in 46 (12.7%) cases. The margins of the conization specimen were free in 266 (84.0%) cases and affected in 50 (16.0%).

During the follow-up period, 55 (13.5%) patients of the total presented an abnormal cervical result of any grade; among them, 7 (1.7%) were high grade. Of the 45 patients treated with laser vaporization, 7 (15.5%) low-grade recurrences and no high-grade recurrence were detected.

During the follow-up period, a total of 986 mRNA detection tests were conducted. The results of these tests are presented in Table 1 and Table 2.

During the follow-up, among the 55 patients who had an abnormal result of any grade, 33 (60%) had a positive mRNA test result. Among the 352 patients who had normal results, 49 (13.9%) had a positive mRNA test during follow-up. These differences were statistically significant (*p* < 0.001).

Among the 266 cases with negative margins in the conization specimen, 32 (12%) had abnormal results of any grade during follow-up. Among the 50 cases with positive margins, 11 (22%) were pathologic results, these differences were not statistically significant (*p* = 0.064).

We observed 7 high-grade recurrences; a total of 3 (1.1%) were diagnosed among the 266 patients with negative margins, and 4 (8%) among the 50 patients with positive margins. These four cases had positive mRNA testing. Considering the 17 patients with positive margins and positive mRNA test, only 4 had high-grade recurrence.

The mRNA test validity measures are shown in Table 3. Due to the absence of high-grade recurrences in the fourth, fifth, and sixth control visits, the validity and safety of the test could not be determined for these clinical visits.

## 4. Discussion

The sensitivity of the mRNA test for detecting cervical dysplasia of any grade was low in our study, with values of 41% at the first control visit, 56% at the second, and 67% at the third. This observation could be attributed to this test type being designed primarily to detect high-grade lesions. Since low-grade lesions express a lower number of oncogenes, the accuracy of these tests decreases for the latter.

Several studies, using different mRNA detection assays, have consistently shown a correlation between the expression level of this molecule and the grade of the lesion to be detected [8,9]. This principle justifies the improved diagnostic yield values observed in our study when the analysis is restricted to high-grade lesions, reaching 70% at the first control visit and 100% at the second and third control visits.

The objective of the follow-up of patients treated for HSIL/CIN is to rule out new or persistent precancerous lesions. A diagnostic test with high sensitivity and NPV should be available [10]. The test evaluated in the present study presents these characteristics. In the first clinical visit, only one HSIL case was detected by cytology, but in the second and third visits, all high-grade recurrences were detected by the mRNA test.

Tropé et al. [10] reported a sensitivity of 45.5% (95% CI: 26.8–65.5) and a negative predictive value (NPV) of 96.2% (95% CI: 93.5–97.8) for the mRNA test in predicting high-grade CIN at 6 months post-conization. The test utilized detected full-length E6 and E7 mRNA from high-risk HPV types 16, 18, 31, 33, and 45. With a follow-up of 18 months, the authors suggested that the low sensitivity of the mRNA test could be attributed to the low expression of oncogenes in new infections.

In an independent study of 143 cases with a mean follow-up of 3.6 years, Persson M et al. [11] reported a sensitivity of 57.1% (95% CI: 25.0–84.2) and a specificity of 93.4% (95% CI: 87.9–96.5). It is relevant to note that this study used the same mRNA test as our investigation, and the analysis was performed on liquid-based samples previously utilized for HPV DNA analysis. The authors attributed the false-negative rate for HPV E6 and E7 to three main factors: the distribution of HR-HPV types in the stored samples, the quality of these stored samples, and the time interval between sample collection and testing (between 6 and 12 months).

In a larger study, Frega A et al. [12] analyzed 475 patients treated by conization between 2003 and 2010. This study reported a higher sensitivity of 73.5% and a negative predictive value (NPV) of 97% for the detection of recurrences. The test for E6 and E7 mRNA from HPV types 16, 18, 31, 33, and 45 was based on a qualitative nucleic acid real-time amplification technique (NASBA).

The overall recurrence rate in this study was 21%. A key aspect of their methodology was the differentiation between residual and recurrent disease to assess the test’s predictive ability specifically for recurrence. Recurrence was defined as the appearance of LSIL or HSIL after negative colposcopy and cytology at 3 and 6 months post-conization. In their conclusions, the authors emphasized that the HPV mRNA test possesses higher sensitivity and a superior NPV for predicting recurrent disease, justifying its implementation in the follow-up of patients with HSIL/CIN treated by conization.

Our study’s findings (showing high sensitivity and high negative predictive value), while subject to the limitations of its retrospective design, demonstrate notable consistency. We followed 407 patients who underwent a total of 986 mRNA detection tests, providing solid information about the follow up.

Considering these results, and anticipating a future with widespread HPV vaccination, we suggest that the mRNA detection test could be a valuable tool in the follow-up of patients treated for high-grade cervical intraepithelial neoplasia (CIN).

From our findings we would like to propose a tiered follow-up strategy:-First Review (to rule out persistence): We recommend that the initial post-treatment review include the mRNA detection test, cytology, and colposcopy.-Subsequent Clinical Controls (to rule out recurrence): If all initial tests are negative, subsequent controls could be limited solely to the mRNA detection test.

This strategy could lead to significant savings by reducing the need for unnecessary cytology and colposcopies in routine follow-up.

Previous screening studies support our conclusions; the HR-HPV E6 and E7 mRNA test used in our center has shown similar sensitivity and slightly higher specificity for the detection of HSIL/CIN compared to the standard HPV DNA-based test [13,14,15]. However, we acknowledge the need for prospective studies to fully validate these recommendations and refine the follow-up strategy.

While acknowledging our study’s retrospective design limitations, our findings indicate high specificity but a low positive predictive value (PPV) for the mRNA test in monitoring patients treated for high-grade squamous intraepithelial lesions (HSIL/CIN). This suggests that a positive mRNA test in our cohort may not reliably predict HSIL/CIN recurrences.

However, it is crucial to consider that the high-grade recurrence rate in our study is low, at 1.72%, compared to rates reported in other investigations [10,11]. This low prevalence of recurrences could explain the low PPV observed in our study. In contrast, the study by Zappacosta R. et al. [16], which included 116 patients treated with conization between 2008 and 2010, reported a significantly higher recurrence rate of 8.6%. In that study, the HPV mRNA test (based on real-time amplification of E6 and E7 from the five most frequent HPV types) showed 100% specificity and 100% PPV, surpassing values obtained with the combination of cytology and the HPV DNA test. The authors of that study concluded that integrating the mRNA test into the follow-up protocol could predict the risk of recurrence after conization earlier, which, in turn, would reduce overtreatment, especially in women over 30 years old.

Our results, along with the demonstrated association between the affected margin of the conization specimen and the mRNA test, suggest that closer monitoring should be performed for patients with a positive mRNA test, even if the detected PPV is low.

Our results indicated a higher likelihood of any-grade recurrence when margins were affected, though this did not reach statistical significance. This observation, while not statistically significant in our study, aligns with existing evidence; for instance, the meta-analysis by Arbyn M et al. had already demonstrated a significantly higher risk of residual or recurrent HSIL/CIN when the margins of the conization specimen were affected [17].

Regarding high-grade recurrence, the same trend was observed, though, once again, it did not reach statistical significance. A notable finding from our results is that four of the seven diagnosed high-grade recurrences had an affected margin in the conization specimen and a positive mRNA test. This suggests a possible relationship between an affected margin, disease persistence, and a positive mRNA test during follow-up.

Recurrence rates of any grade were similar regardless of the treatment performed, whether it was laser vaporization or conization. In the group of patients treated with laser vaporization, no high-grade recurrence was detected. This finding could be attributed to patient selection, as most cases involved small lesions, classified as CIN 2, and with a type 1 transformation zone.

Similarly, the absence of high-grade recurrences in the group of patients without residual lesion in the conization specimen might be justified by the small extent of the initial lesion, which was likely completely excised during the diagnostic biopsy.

In our study, the observed recurrence rate was remarkably low, with only 7 out of 407 cases (1.72%) experiencing recurrence. This finding significantly contrasts with rates reported in other previous investigations [6,7]. We hypothesize that this lower incidence of recurrences may be associated with post-treatment human papillomavirus (HPV) vaccination in patients diagnosed with high-grade cervical intraepithelial neoplasia (CIN).

At our unit, we follow a standardized protocol where HPV vaccination is prescribed to all patients treated for high-grade squamous intraepithelial lesions (HSIL)/CIN who have not been previously immunized. It is important to note that this vaccination is provided free of charge through primary care centers.

These results are consistent with the conclusions of recent meta-analyses. Specifically, studies by Lichter K and Kechagias KS, published in 2020 and 2022, respectively, suggest that adjunctive HPV vaccination in the context of surgical excision for CIN 2 or 3 is associated with a reduced risk of CIN recurrence [18,19].

## 5. Conclusions

Our findings reinforce the need to implement systematic human papillomavirus (HPV) vaccination programs for non-immunized patients who have undergone treatment for high-grade cervical intraepithelial neoplasia (high-grade CIN). Evidence suggests that this strategy significantly reduces disease recurrence rates.

In the follow-up of human papillomavirus (HPV)-immunized patients treated for high-grade cervical intraepithelial neoplasia (HSIL/CIN), the mRNA detection test demonstrates high sensitivity and a high negative predictive value (NPV) for identifying persistent disease.

Our findings suggest a tiered monitoring protocol; at the first post-treatment visit, a combination of HPV mRNA testing and cytology is recommended. For subsequent visits, HPV mRNA testing alone appears sufficient; a negative result would indicate no need for further testing.

It is crucial to maintain close surveillance for patients presenting with positive margins on the conization specimen and a positive mRNA test result. Finally, we observed that recurrence rates are comparable between conization and laser vaporization therapies, suggesting similar efficacy in disease control.

## Figures and Tables

**Table 1 vaccines-13-00823-t001:** Results of mRNA testing during the follow-up clinical visits and the cases of recurrence.

Follow-Up Clinical Visits	First N = 424	Second N = 357	Third N = 182	Fourth N = 97	Fifth N = 16	Sixth N = 6
Negative mRNA	355 (89.9%)	311 (91.2%)	141 (83.4%)	48 (78.7%)	11 (78.6%)	5 (83.3%)
Positive mRNA	40 (10.1%)	30 (8.8%)	28 (16.6%)	13 (21.3%)	3 (21.4%)	1 (16.7%)
Any Grade Recurrence	26 (6.6%)	15 (4.4%)	10 (6%)	6 (10%)	2 (14.3%)	0 (0%)
High Grade Recurrence	3 (0.6%)	1 (0.3%)	3 (1.8%)	0 (0%)	0 (0%)	0 (0%)

Acronyms: N—number of cases.

**Table 2 vaccines-13-00823-t002:** Recurrences with positive mRNA test according to treatment type and outcome.

	Positive mRNA	Any Grade Recurrence	High Grade Recurrence	Total Number of Cases
Treatment: - Laser vaporization - Conization	9 (20%) 73 (20.1%)	7 (15.5%) 48 (13.2%)	0 (0%) 7 (1.9%)	45 362
Pathology after conization: - No residual lesion - HSIL/CIN	6 (13%) 67 (21.2%)	5 (10.8%) 43 (13.6%)	0 (0%) 7 (2.2%)	46 316
Margins after conization: - Free - Affected	50 (18.7) 17 (34%)	32 (12%) 11 (22%)	3 (1.1%) 4 (8%)	266 50

Acronyms: HSIL/CIN—high-grade cervical intraepithelial neoplasia.

**Table 3 vaccines-13-00823-t003:** mRNA test validity measures.

	**First FU Clinical Visit**		**Second FU Clinical Visit**		**Third FU Clinical Visit**	
	**Any Grade**	**High Grade**	**Any Grade**	**High Grade**	**Any Grade**	**High Grade**
Sensitivity % (95% CI)	41 (26–55)	88 (65–110)	56 (37–75)	100 (100–100)	67 (40–93)	100 (100–100)
Specificity % (95% CI)	94 (91–96)	91 (89–94)	95 (93–97)	92 (89–95)	87 (82–92)	85 (80–91)
PPV % (95% CI)	45 (30–60)	18 (6–29)	47 (29–65)	10 (2–16)	29 (12–45)	14 (1–27)
PNV % (95% CI)	93 (90–95)	100 (100–100)	100 (94–99)	100 (100–100)	97 (94–100)	100 (100–100)

Acronyms: PPV—positive predictive value; NPV—negative predictive value; CI—confidence interval; n—number of cases; FU—follow-up.

## Data Availability

This study is based on real-world patient data, including demographics and comorbidity factors, that cannot be communicated due to patient privacy concerns.

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
