# Peer review of "HPV E6/E7 mRNA Testing in the Follow-Up of HPV-Vaccinated Patients After Treatment for High-Grade Cervical Intraepithelial Neoplasia"

_vaccines, 2025, doi:10.3390/vaccines13080823_

Round 1

Reviewer 1 Report

Comments and Suggestions for Authors

This study investigated the clinical utility of E6/E7 mRNA detection in the follow-up of high-grade cervical intraepithelial neoplasia (HSIL/CIN) in non vaccinated HPV patients after treatment, which has certain clinical significance. However, there are some problems that need be solved.

  1. The study only focuses on E6/E7 mRNA detection, but lacks direct comparative data with traditional HPV DNA detection (such as parallel sensitivity, specificity, and other indicators). Although the article mentions that HPV DNA testing has low specificity, the performance differences between the two methods were not directly compared in this cohort, making it difficult to highlight the clinical advantages of mRNA testing.
  2. The study included 6 patients (2.6%) who had not been histologically diagnosed, which may affect the accuracy of the results. In addition, the lesion characteristics of patients treated with laser vaporization therapy (such as lesion size and transformation zone type) were not detailed, leading to confounding factors in the comparison of different treatment methods (cone cutting vs laser).
  3. The first follow-up time for patients with positive margins is 4 months, while for others it is 6 months, which may result in inconsistent testing window periods. In addition, the follow-up time points (18, 42, 78 months) were unevenly spaced and lacked standardized follow-up plans, which affected the time-dependent analysis of the results.

Author Response

Comment 1: The study only focuses on E6/E7 mRNA detection, but lacks direct comparative data with traditional HPV DNA detection (such as parallel sensitivity, specificity, and other indicators). Although the article mentions that HPV DNA testing has low specificity, the performance differences between the two methods were not directly compared in this cohort, making it difficult to highlight the clinical advantages of mRNA testing.

Response 1: Indeed, a direct comparison between the performance of the E6/E7 mRNA detection test and the HPV DNA detection assay was not feasible. This is due to the fact that, as of 2015, our center exclusively adopted the mRNA detection test for the follow-up of patients treated for high-grade cervical intraepithelial neoplasia (CIN). Consequently, we do not possess parallel data obtained through both methodologies within the same patient cohort.

Comment 2: The study included 6 patients (2.6%) who had not been histologically diagnosed, which may affect the accuracy of the results. In addition, the lesion characteristics of patients treated with laser vaporization therapy (such as lesion size and transformation zone type) were not detailed, leading to confounding factors in the comparison of different treatment methods (cone cutting vs laser).

Response 2: We acknowledge that the initial phrasing of this point was not as precise as intended. It should state that six patients (2.6%) were included who did not have a histological diagnosis prior to the intervention but presented with HSIL cytology, and a high-grade lesion was subsequently confirmed in the conization specimen.

The decision between conization and laser vaporization is made on an individualized basis, informed by a detailed clinical evaluation encompassing cytological, colposcopic, and histological findings, as well as the patient's characteristics and preferences. The overarching goal is always to achieve effective lesion eradication and optimize long-term quality of life. Further clarification on this decision-making process has been added to the Materials and Methods section.

At our center, loop electrosurgical excision procedure (LEEP/LLETZ) is the standard treatment for high-grade cervical lesions (CIN2/CIN3). This method allows for tissue removal and subsequent histological analysis. However, to minimize potential side effects, we perform laser vaporization in selected cases. This ablative technique is reserved for: (a) Young patients who have completed their reproductive desires; (b) Lesions that are completely visible via colposcopy; (c) Cases where there's no suspicion of invasive disease, as confirmed by prior biopsies.

Comment 3: The first follow-up time for patients with positive margins is 4 months, while for others it is 6 months, which may result in inconsistent testing window periods. In addition, the follow-up time points (18, 42, 78 months) were unevenly spaced and lacked standardized follow-up plans, which affected the time-dependent analysis of the results.

Response 3: You've correctly identified a potential inconsistency. According to the 2015 clinical guidelines, the initial follow-up for patients with positive margins was indeed at 4 months, while for those with clear margins, it was at 6 months. This, most likely, introduced a two-month variation in the timing of subsequent reviews.

Regarding the follow-up periods, we will make a clarification in the revised text. The 2015 follow-up protocol for patients treated for high-grade CIN consisted of a first control (at 4 months for positive margins, and 6 months for clear margins), followed by three annual controls, and thereafter, biennial reviews.

Reviewer 2 Report

Comments and Suggestions for Authors

In principle it is an interesting research, not new given that others had tried to address same issue. There are few issues in your research design, which concern me given your aims. 1.it’s a retrospective study, which brings with it many bias, including patient and treatment bias and therefore outcome. It’s unclear why one procedure was chosen over the other, laser vs conization, what was the criteria used to use either ? Specimens with clear margins or not? Why did you include samples with residual disease? This could, by itself, influenced the outcome. 2. The English throughout the manuscript is difficult to understand. There are paragraphs l, which need to be rewritten and simplified specifically under Results and Discussion. 3. We used tables so we don’t have to repeat the same material in the text. In your tablets and text you have to used the same terms description ( VPP,  VPN or PPV, PNV). Also, as it concerns the number and title goes on the top and any description on the contents should be in the bottom. Remember to keep it simple so readers can clearly understand your results, and make sense of your conclusions. 4. Given the differences in study designs, it will be difficult to clearly and unbiased determine the validity of the sensitivity tests used to diagnose recurrence. This is especially important when you compare the results of Frega A et al and every other study you reviewed, including yours. 5. Under Discussion paragraph from 182-down needs to be rewritten. It’s extremely difficult to follow, you definitely need professional editing. Under discussion do not repeat results, outline any relevant finding only if you want to make a statement about how it compares or not to similar studies. Again, keep in mind the source of your data when comparing to others. 6. What do you mean by revisions ( under material and methods).Please explain. Do you mean Follow up visits?? 

Comments on the Quality of English Language

Very difficult to follow the whole text. It needs to be edited by a professional. Terms definition should be the same across the entire text. Tables are use to describe findings in a simple way . Describing same results in text and tables is not appropriate. 

Author Response

Comment 1: it’s a retrospective study, which brings with it many bias, including patient and treatment bias and therefore outcome. It’s unclear why one procedure was chosen over the other, laser vs conization, what was the criteria used to use either ? Specimens with clear margins or not? Why did you include samples with residual disease? This could, by itself, influenced the outcome

Response 1: You've raised highly relevant observations regarding the inherent limitations of a retrospective study and the potential for bias. We acknowledge that the retrospective nature of our study introduces variables beyond our direct control, including potential biases in patient and treatment selection that could influence the observed outcomes.

Regarding the choice between laser vaporization and conization, we understand the need for clarity. The decision was not arbitrary; it was based on specific clinical criteria that we have clarified and added to the Materials and Methods section of our manuscript. In summary, loop electrosurgical excision procedure (LEEP/LLETZ) conization remains our standard treatment for high-grade cervical intraepithelial neoplasia (CIN2/CIN3), as it allows for tissue excision and histological analysis. However, laser vaporization was employed in selected cases to minimize potential sequelae, specifically in: (a) Young patients who have completed their reproductive desires; (b) Lesions that are completely visible via colposcopy; (c) Cases where there's no suspicion of invasive disease, as confirmed by prior biopsies.

This individualized approach aims to effectively treat the lesion while optimizing patient outcomes. The decision to include patients with affected margins after conization in the follow-up aimed to determine if the presence of positive surgical margins increases the likelihood of lesion recurrence. Our findings showed a difference in the recurrence rate between groups with and without affected margins, although this difference did not reach statistical significance. While this result doesn't establish a statistically robust causality or association, it provides valuable information for guiding rigorous follow-up in these patients.

Comment 2: The English throughout the manuscript is difficult to understand. There are paragraphs l, which need to be rewritten and simplified specifically under Results and Discussion.

Response 2: Thank you very much for your detailed feedback on the Discussion section. We fully understood that the paragraph starting at line 182 lacked the necessary clarity and required a significant rewrite. We took good note of your recommendation to improve its flow and, consequently, its comprehension by the reader.

Furthermore, we completely agreed on the importance of avoiding the repetition of results in the Discussion section. Following your guidance, we conducted a thorough revision of almost the entire manuscript, ensuring we addressed this point and optimized the presentation of our findings in relation to existing literature.

Comment 3: We used tables so we don’t have to repeat the same material in the text. In your tablets and text you have to used the same terms description ( VPP,  VPN or PPV, PNV). Also, as it concerns the number and title goes on the top and any description on the contents should be in the bottom. Remember to keep it simple so readers can clearly understand your results, and make sense of your conclusions

Response 3: In response to your feedback, we have made the following adjustments:

We've thoroughly reviewed both the text and tables to ensure consistent use of terminology. We've adjusted the format of all tables to meet your guidelines. The number and title of each table are now at the top, and any descriptive notes or content explanations have been moved to the bottom. We've also revised the general wording to ensure the results are presented as clearly and concisely as possible, making the findings and conclusions easier for all readers to understand.

We believe these changes significantly improve the clarity and presentation of our results, as you suggested.

Comment 4: Given the differences in study designs, it will be difficult to clearly and unbiased determine the validity of the sensitivity tests used to diagnose recurrence. This is especially important when you compare the results of Frega A et al and every other study you reviewed, including yours

Response 4: Thank you very much for your comment. We agree that the comparison between different studies sometime can be tricky, but this is something that we cannot control, and the reader should critically compare each result to obtain their own conclusion.

Comment 5: Under Discussion paragraph from 182-down needs to be rewritten. It’s extremely difficult to follow, you definitely need professional editing. Under discussion do not repeat results, outline any relevant finding only if you want to make a statement about how it compares or not to similar studies. Again, keep in mind the source of your data when comparing to others.

Response 5: Thank you very much for your detailed feedback on the Discussion section. We fully understand that the paragraph starting at line 182 lacks the necessary clarity and requires a significant rewrite. We've taken good note of your recommendation to improve its flow and, consequently, its comprehension by the reader.

Furthermore, we completely agree on the importance of avoiding the repetition of results in the Discussion section. Following your guidance, we've conducted a thorough revision of almost the entire manuscript, ensuring we address this point and optimize the presentation of our findings in relation to existing literature.

Comment 6: What do you mean by revisions ( under material and methods).Please explain. Do you mean Follow up visits?? 

Response 6: Yes, we are indeed referring to follow-up visits. We will make the corresponding correction in the text.

Comment 7: Comments on the Quality of English Language. Very difficult to follow the whole text. It needs to be edited by a professional. Terms definition should be the same across the entire text. Tables are use to describe findings in a simple way . Describing same results in text and tables is not appropriate.

Response 7: Thank you very much for your direct feedback on the quality of the language. We understand the text is difficult to follow and we recognize the importance of clear, professional writing.

We underwent an extensive English review to ensure consistent terminology throughout the text. Additionally, we've already revised the presentation of our findings to avoid any unnecessary repetition between the text and tables, making sure the information is presented as simply and effectively as possible. We're committed to making the necessary improvements so the document is easy to follow.

Round 2

Reviewer 1 Report

Comments and Suggestions for Authors

This revised manuscript can be accepted.

Author Response

Dear Reviewer,

Thank you very much again for your kind comments.

Best regards.

I. Zapardiel.

Reviewer 2 Report

Comments and Suggestions for Authors

Thanks for addressing my concerns. Still few questions. Introduction. Could you explain the wide (5%-30%) risk of persistence or recurrence among non-vaccinated patients. Obviously, there is not consistence in published data  

2. Throughout the text we need consistency in terms use: control visits?, number of revisions?, follow ups?  Do they all mean the same? Then choose the appropriate term.

3. Could you rewrite the paragraph from 97-99. Not clear if treatment failure was determined by using cytology and/or biopsy?

4. Results: what exactly you mean by 297 (73%) CIN 2, and 104 ( 25.6%) of CIN? Please rewrite Lane 121 for consistency.

5. Tablet 2 contains same information written in paragraph bellow it, is it needed?

6. Lane 173 is review same as follow up visit? Still need a bit of editing from lane 166-206.

7. Lane 214, I suggest to write: from our findings we would like to propose a tiered follow up strategy. 

Comments on the Quality of English Language

Still needs some work. Last tense, semicolon use, and etc 

Author Response

Comment 1: Introduction. Could you explain the wide (5%-30%) risk of persistence or recurrence among non-vaccinated patients. Obviously, there is not consistence in published data 

Response 1: Indeed, the lack of consistency in published data is noteworthy. This variability could be attributed to several key factors such as heterogeneity in diagnostic and follow-up criteria used in each study as well as in definition of persistence or recurrence disease. There are significant differences in the frequency of evaluations, the detection methods used (such as viral tests, cytology, or histology), the threshold for considering a case as persistent or recurrent, and the patient follow-up duration among publications. These methodological inconsistencies directly contribute to the disparity in results.

Comment 2: Throughout the text we need consistency in terms use: control visits?, number of revisions?, follow ups? Do they all mean the same? Then choose the appropriate term.

Response 2: Dear reviewer, We have replaced the term 'revisions' by 'control visits' throughout the manuscript. You are right it could be confusing. Thank you for the comment.

Comment 3: Could you rewrite the paragraph from 97-99. Not clear if treatment failure was determined by using cytology and/or biopsy?

Response 3: Our intention was to clarify that the cytological diagnosis was histologically confirmed by biopsy. The paragraph has been modified accordingly.

Comment 4: Results: what exactly you mean by 297 (73%) CIN 2, and 104 ( 25.6%) of CIN? Please rewrite Lane 121 for consistency.

Response 4: We've modified the text to optimize its clarity and comprehensibility.

Comment 5: Table 2 contains same information written in paragraph bellow it, is it needed?

Response 5: Thank you for the comment. We think the text clarify the content of Table 2. Even if some information is repeated, we think the information contained in the paragraph completes and helps to a better understanding of the whole content.

Comment 6: Lane 173 is review same as follow up visit? Still need a bit of editing from lane 166-206.

Response 6: We have taken into account your comment. We revised the text between lanes 166 and 206, implementing modifications to improve the clarity and consistency. Thank you very much for pointing it out.

Comment 7: Lane 214, I suggest to write: from our findings we would like to propose a tiered follow up strategy.

Response 7: Absolutely. We have implemented the suggested change.

Comment 8: Comments on the Quality of English Language. Still needs some work. Last tense, semicolon use, and etc

Response 8: We have tried to improve the language, we hope now it can be considered for publication. Thanks a lot for all the review that for sure has improved the manuscript.